# CarD uses a minor groove wedge mechanism to stabilize the RNA polymerase open promoter complex

**Brian Bae, James Chen, Elizabeth Davis, Katherine Leon, Seth A Darst\*, Elizabeth A Campbell\***

Laboratory for Molecular Biophysics, The Rockefeller University, New York, United States

**Abstract** A key point to regulate gene expression is at transcription initiation, and activators play a major role. CarD, an essential activator in *Mycobacterium tuberculosis*, is found in many bacteria, including *Thermus* species, but absent in *Escherichia coli*. To delineate the molecular mechanism of CarD, we determined crystal structures of *Thermus* transcription initiation complexes containing CarD. The structures show CarD interacts with the unique DNA topology presented by the upstream double-stranded/single-stranded DNA junction of the transcription bubble. We confirm that our structures correspond to functional activation complexes, and extend our understanding of the role of a conserved CarD Trp residue that serves as a minor groove wedge, preventing collapse of the transcription bubble to stabilize the transcription initiation complex. Unlike *E. coli* RNAP, many bacterial RNAPs form unstable promoter complexes, explaining the need for CarD.

## Introduction

Decades of research using *Escherichia coli* (*Eco*) as a model system inform most of our understanding of how bacteria control transcription initiation. First, dissociable promoter specificity subunits, σ factors, direct the catalytic core of the RNA polymerase (RNAP) to promoter DNA sites and play a key role in unwinding the DNA duplex to create the transcription bubble in the RNAP holoenzyme open promoter complex (RPo) (*Feklistov et al., 2014*). Second, DNA-binding transcription factors either activate or repress the initiation rate (*Browning and Busby, 2004*).

The majority of transcription activators characterized to date are dimeric proteins that bind operators upstream of the promoter −35 element and directly contact the RNAP α subunit (*Ebright, 1993*), the $\sigma_4$ domain positioned at the −35 element, or both (*Nickels et al., 2002*; *Dove et al., 2003*; *Jain et al., 2004*). Activators can accelerate initiation by stabilizing the initial RNAP/promoter complex, by stimulating the isomerization of the initial RNAP/promoter complex to RPo (i.e., unwinding the duplex DNA to form the transcription bubble), or both (*Li et al., 1997*; *Roy et al., 1998*).

CarD, first identified as a regulator of ribosomal RNA (rRNA) transcription in *Mycobacterium tuberculosis* (*Mtb*), is a transcriptional activator widely distributed among bacterial species, including *Thermus* species (*Stallings et al., 2009*; *Srivastava et al., 2013*), but is absent in *Eco* (*Table 1*). CarD is a global regulator (*Srivastava et al., 2013*) that is an essential protein in *Mtb* (*Stallings et al., 2009*), the causative agent of tuberculosis. A deeper understanding of the CarD functional mechanism and its role in the *Mtb* transcription program is therefore warranted.

Crystal structures of *Tth* (*Srivastava et al., 2013*) and *Mtb* (*Gulten and Sacchettini, 2013*) CarD reveal an N-terminal domain with a Tudor-like fold (CarD-RID, RNAP interacting domain) in common with the *Eco* transcription repair coupling factor (TRCF)-RID (*Deaconescu et al., 2006*; *Stallings et al., 2009*; *Weiss et al., 2012*), and a helical C-terminal domain (CarD-CTD). Unique among known

**\*For correspondence:** darst@ rockefeller.edu (SAD); elizabeth. campbell0@gmail.com (EAC)

**Competing interests:** The authors declare that no competing interests exist.

**eLife digest** Inside cells, molecules of double-stranded DNA encode the instructions needed to make proteins. To make a protein, the two strands of DNA that make up a gene are separated and one strand acts as a template to make molecules of messenger ribonucleic acid (or mRNA for short). This process is called transcription. The mRNA is then used as a template to assemble the protein. An enzyme called RNA polymerase carries out transcription and is found in all cells ranging from bacteria to humans and other animals.

Bacteria have the simplest form of RNA polymerase and provide an excellent system to study how it controls transcription. It is made up of several proteins that work together to make RNA using DNA as a template. However, it requires the help of another protein called sigma factor to direct it to regions of DNA called promoters, which are just before the start of the gene. When RNA polymerase and the sigma factor interact the resulting group of proteins is known as the RNA polymerase 'holoenzyme'.

Transcription takes place in several stages. To start with, the RNA polymerase holoenzyme locates and binds to promoter DNA. Next, it separates the two strands of DNA and exposes a portion of the template strand. At this point, the DNA and the holoenzyme are said to be in an 'open promoter complex' and the section of promoter DNA that is within it is known as a 'transcription bubble'. Another protein called CarD helps to speed up transcription but it is not clear how this stage of the process works.

Bae et al. have now used X-ray crystallography to reveal the structure of CarD bound to the RNA polymerase holoenzyme and a DNA promoter. The structures show that one part of CarD interacts with the DNA at the start of the transcription bubble, and another part binds to the RNA polymerase. CarD fits between the two strands of DNA in the promoter, like a wedge, to keep the strands apart. Therefore, CarD stabilizes the open promoter complex and prevents the transcription bubble from collapsing.

These findings reveal a previously unseen mechanism involved in activating transcription and will guide further experiments probing the role of CarD in living cells. Another study by Bae, Feklistov et al.—which involves some of the same researchers as this study—reveals that the sigma factor also binds to DNA at the start of the transcription bubble. The general principles outlined by these studies may help to identify other proteins that regulate transcription.

transcription activators, the CarD-RID interacts with the RNAP β subunit β1-lobe (*Stallings et al., 2009*; *Weiss et al., 2012*) (corresponding to the eukaryotic RNAP II Rpb2 protrusion domain; *Cramer et al., 2001*), which is near the upstream portion of the transcription bubble in RPo (*Bae et al., 2015*). The disposition of the CarD-CTD with respect to the CarD-RID is widely divergent in *Tth* and *Mtb* CarD crystal structures, leading to conflicting models for the CarD activation mechanism (*Gulten and Sacchettini, 2013*; *Srivastava et al., 2013*). To resolve these ambiguities, we determined crystal structures of *Thermus aquaticus* (*Taq*) transcription initiation complexes (RPo) (*Bae et al., 2015*) containing CarD (*Figure 1A,B*). The structures show that CarD interacts with the unique DNA topology of the upstream double-stranded/single-stranded (ds/ss) DNA junction of the transcription bubble. Additional biochemical data confirm that our structures correspond to functional activation complexes, and extend our understanding of the role of a universally conserved CarD Trp residue in stabilizing the unwound transcription bubble, thereby stabilizing the transcription initiation complex.

Throughout this work, we use three promoter sequences, full con (*Gaal et al., 2001*), *Tth* 23S (*Hartmann et al., 1987*), and *Mtb* AP3 (*Gonzalez-y-Merchand et al., 1996*) (*Figure 1—figure supplement 1*). The full con promoter sequence, derived by in vitro evolution, is likely to be optimized for binding to Eσ$^A$. We use this sequence only for structural studies where high-affinity, homogeneous complexes are critical for crystallization. AP3 is a native *Mtb* rRNA promoter and its regulation by *Mtb* CarD has been well characterized (*Srivastava et al., 2013*; *Davis et al., 2015*). In order to biochemically characterize more than one promoter, we also studied 23S, a native *Tth* rRNA promoter. In promoter-based assays, the effects of *Tth* or *Mtb* CarD on each promoter were qualitatively the same. In general, we present the results from *Tth* 23S since most of the studies used *Thermus* Eσ$^A$ and CarD. In some cases, it was advantageous to use *Mtb* AP3 instead and we note the rationales below.

**Table 1**. Distribution of CarD in bacterial phyla

| Phyla* | Clades and colloquial names noted. Select genera within some phyla are also listed | CarD presence in phyla | # of completed genomes and draft assemblies† |
|---|---|---|---|
| **Acidobacteria/ Fibrobacter** | **diderm Gram−** | **Yes (only Acidobacteria)** | **24** |
| **Actinobacteria** | **monoderm, high G + C Gram+: *Streptomyces, Mycobacteria*** | **Yes** | **932** |
| **Aquificae** | **diderm Gram−: glidobacteria** | **Yes** | **16** |
| Bacteroidetes | diderm Gram−: Green sulfur bacteria | No | 468 |
| Caldiserica | diderm Gram− | No | 2 |
| **Chlamydiae** | **diderm Gram− Planctobacteria: *Chlamydia trachomatis*** | **Yes** | **21** |
| Chlorobi | diderm Gram− | No | 12 |
| Chloroflexi | diderm Gram−: glydobacteria | No | 32 |
| Chrysiogenetes | diderm Gram−: *Desulfurispirillum* | No | 2 |
| **Cyanobacteria‡** | **diderm Gram−: glydobacteria** | **Yes** | **103** |
| Deferribacteres | diderm Gram− | No | 6 |
| **Deinococcus–Thermus** | **diderm Gram−: glydobacteria** | **Yes** | **43** |
| Dictyoglomi | diderm Gram− | No | 2 |
| Elusimicrobia | diderm Gram− | No | 3 |
| **Firmicutes** | **monoderm low G + C Gram+: *Bacillus, Clostridium*** | **Yes** | **1149** |
| Fusobacteria | diderm Gram− | No | 25 |
| Gemmatimonadetes | diderm Gram− | No | 5 |
| Lentisphaerae | diderm Gram− | No | 2 |
| Nitrospirae | diderm Gram− | No | 10 |
| Planctomycetes | diderm Gram−: planctobacteria | No | 22 |
| **Proteobacteria-α** | **diderm Gram−: *Rickettsia, Rhizobium*** | **Yes** | **678** |
| Proteobacteria-β | diderm Gram−: *Bordetella, Neisseria* | No | 350 |
| Proteobacteria-γ | diderm Gram−: *Escherichia, Pseudomonas* | No | 982 |
| **Proteobacteria-δ** | **diderm Gram−: *Desulfovibrio, Geobacter*** | **Yes** | **142** |
| Proteobacteria-ε | diderm Gram−: *Helicobacter* | No | 78 |
| **Spirochaetes** | **di-derm Gram−: *Borrelia, Treponema*** | **Yes** | **81** |
| Synergistetes | diderm Gram− | No | 18 |
| Tenericutes | Monoderm: *Mycoplasma* | No | 132 |
| **Thermodesulfobacteria** | **diderm Gram−: glidobacteria** | **Yes** | **3** |
| Thermotogae | diderm Gram− | No | 26 |
| Verrucomicrobia | diderm Gram− | No | 37 |

*Phyla list based on the list of prokaryotic names with standing in nomenclature (LPSN) (http://www.bacterio.net/-classifphyla.html) and the NCBI taxonomy list (http://www.ncbi.nlm.nih.gov/Taxonomy/Browser/wwwtax.cgi). The diverse phylum proteobacteria are divided into subgroups of α, β, γ, δ and ε.

†Genomes and draft assemblies sequenced list are shown to illustrate representation of each phylum in the Blast database and gathered from http://www.ncbi.nlm.nih.gov/genomes/MICROBES/microbial_taxtree.html.

‡Phyla containing CarD are highlighted in bold.

Method: Using the Blast database search engine (http://blast.ncbi.nlm.nih.gov/Blast.cgi?PROGRAM=blastp&PAGE_TYPE=BlastSearch&LINK_LOC=blasthome) we searched for sequences similar to *Tth* CarD with restrictions of amino acid length of 120:200 amino acids within each phylum.

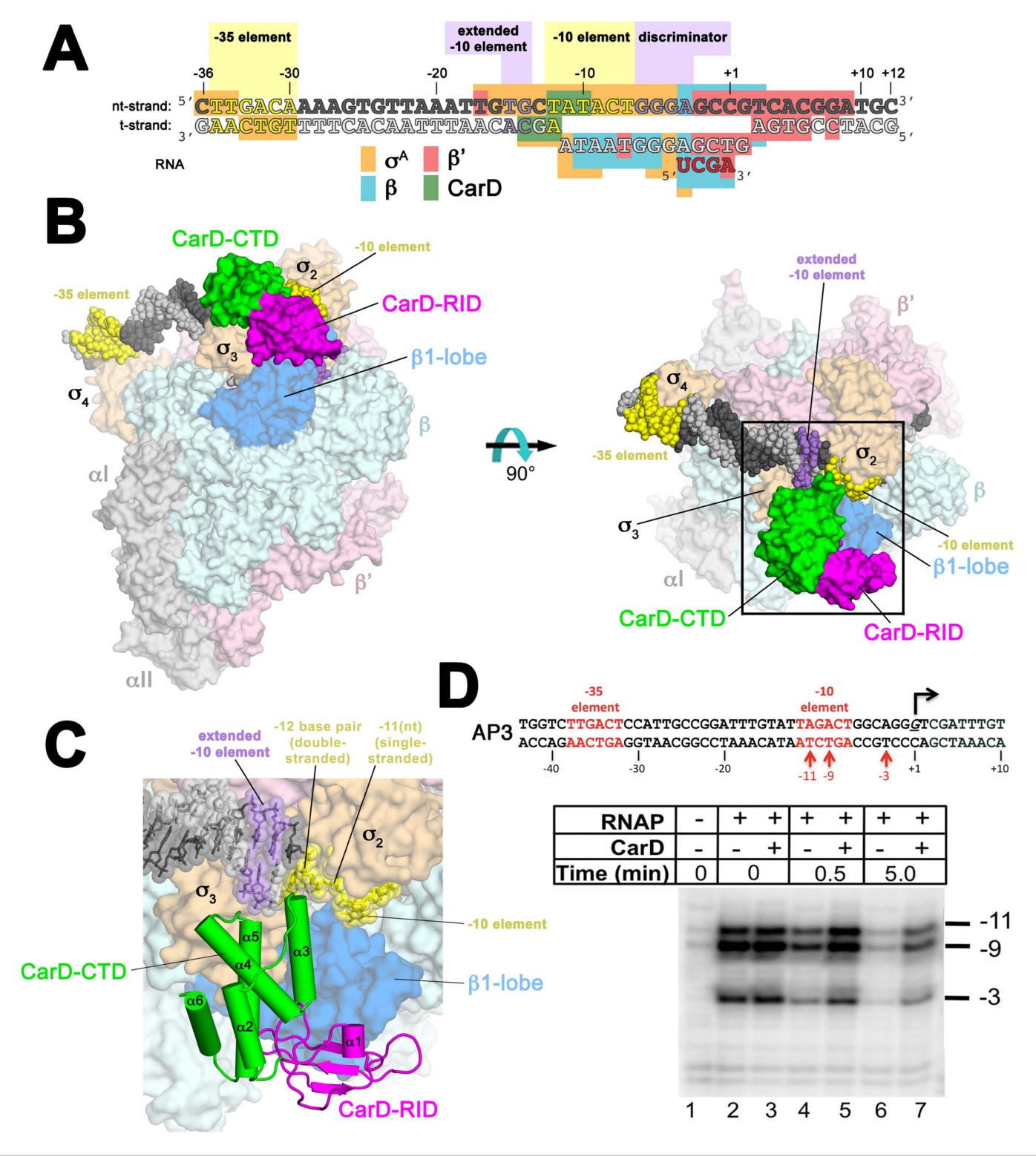

**Figure 1**. Structure of the Thermus CarD/RPo complex. (**A**) Synthetic oligonucleotides used for CarD/RPo crystallization. The numbers above denote the DNA position with respect to the transcription start site (+1). The DNA sequence is derived from the full con promoter (*Gaal et al., 2001*). The −35 and −10 (Pribnow box) elements are shaded yellow, the extended −10 (*Keilty and Rosenberg, 1987*) and discriminator (*Feklistov et al., 2006*; *Haugen et al., 2006*) elements purple. The nt-strand DNA (top strand) is colored dark grey; the t-strand DNA (bottom strand), light grey; the RNA transcript, red. The colored blocks denote protein/nucleic acid interactions: σ^A, orange; β, cyan; β′, pink; CarD, green. CarD interacts exclusively at the upstream junction of the

*Figure 1. continued on next page*

*Figure 1. Continued*

transcription bubble. (**B**) Overall structure of CarD/RPo—two orthogonal views. The nucleic acids are shown as CPK atoms and color-coded as above. Proteins are shown as molecular surfaces. The RNA polymerase (RNAP) holoenzyme is color coded as follows: αI, αII, ω, grey; β′, light pink; Δ1.1σ$^A$, light orange; β is light cyan except the β1-lobe (interacting with the CarD-RID, corresponding to RNAP β subunit residues 18–138 and 333–392) is light blue. The CarD-RID is magenta, CarD-CTD green. In the right view, the boxed region is magnified in (**C**). (**C**) Magnified view illustrating the CarD-RID/β1-lobe protein/protein interaction and CarD-CTD (α3 and α5)/DNA interactions at the upstream ds(−12)/ss(−11) junction of the transcription bubble. (**D**) CarD does not alter the transcription bubble. KMnO$_4$ footprints (t-strand) of Thermus RNAP holoenzyme on the *Mtb* AP3 promoter. (*Top*) Sequence of the AP3 promoter (*Hartmann et al., 1987*). T-strand thymidines rendered KmnO$_4$ reactive by RNAP are denoted (red arrows). (*Bottom*) KMnO$_4$ footprints. Lane 1, no protein added; lanes 2–3, RNAP holoenzyme − or + CarD (respectively); lanes 4–7, the effect of incubating with a competitor promoter trap for the indicated amounts of time.

The following figure supplements are available for figure 1:

**Figure supplement 1**. Sequences of *Mtb rrnA*AP3 (*Gonzalez-y-Merchand et al., 1996*) and *Tth* 23S ribosomal RNA (rRNA) (*Hartmann et al., 1987*), promoters used in in vitro assays, and full con (*Gaal et al., 2001*) used for structural studies.

**Figure supplement 2**. Crystal packing interactions in CarD/RPo P4$_3$2$_1$2 crystals.

**Figure supplement 3**. CarD/β1-lobe structure.

**Figure supplement 4**. Slight movement of CarD-CTD towards DNA when DNA is present.

**Figure supplement 5**. Data and model quality.

**Figure supplement 6**. CarD does not alter the structure of the transcription bubble.

## Results

### Overall structure of the *Thermus* CarD/RPo complex

Crystals of CarD transcription activation complexes were prepared by soaking *Tth* CarD into *Taq* Δ1.1σ$^A$-holoenzyme/us-fork (−12 bp) or full RPo crystals (*Bae et al., 2015*). Analysis of the diffraction data indicated high occupancy of one CarD molecule bound to each of two RNAP/promoter complexes in the asymmetric unit of the crystal lattice (*Figure 1—figure supplement 2*). Docking CarD onto the RNAP was facilitated by a high-resolution crystal structure of a *Tth* CarD/*Taq* β1-lobe complex (2.4 Å-resolution, *Table 2*, *Figure 1—figure supplement 3*, *Figure 1—figure supplement 4*). The structures of CarD transcription activation complexes were refined to 4.4 and 4.3 Å-resolution, respectively (*Table 2*, *Figure 1—figure supplement 5*). The protein/protein and protein/nucleic acid interactions were essentially identical among all of the four crystallographically independent complexes, so the more complete and higher resolution CarD/RPo structure (*Figure 1A,B*, *Figure 1—figure supplement 5*, *Table 2*) is described here. Although the CarD bound to one RPo in the crystallographic asymmetric unit made crystal-packing interactions with a symmetry-related CarD, the CarD bound to the second RPo did not participate in any crystal-packing interactions (*Figure 1—figure supplement 2*), indicating the architecture and interactions observed here are unlikely to be influenced by crystal packing interactions and likely represent the functional activation complex in solution.

### The CarD-CTD interacts with the upstream ds/ss junction of the transcription bubble

The relative orientation of the CarD domains (CarD-RID, CarD-CTD) seen in the *Thermus* CarD (*Srivastava et al., 2013*) and CarD/β1-lobe (*Figure 1—figure supplement 3*) structures is only slightly altered in the *Thermus* CarD/RPo complex: the CarD-CTD is rotated ~11° (with respect to the CarD-RID) to interact with the DNA (*Figure 1—figure supplement 4*). By maintaining the CarD-RID/CTD interface seen in all the *Tth* CarD structures, binding of the CarD-RID to the RNAP β1-lobe (*Figure 1B,C*, *Figure 1—figure supplement 4*) (*Stallings et al., 2009*; *Weiss et al., 2012*; *Gulten and Sacchettini, 2013*) positions the CarD-CTD to interact directly with the upstream ds/ss junction of the transcription bubble (*Figure 1A–C*).

**Table 2.** Crystallographic statistics

| | Holo-bubble-CarD | Holo-fork-CarD | CarD/β1-lobe |
|---|---|---|---|
| **Data collection** | | | |
| Space group | $P4_32_12$ | $P4_32_12$ | I4 |
| Combined datasets | 4 | 6 | 1 |
| **Cell dimensions** | | | |
| a (Å) | 289.84 | 293.15 | 149.32 |
| b (Å) | 289.84 | 293.15 | 149.32 |
| c (Å) | 536.34 | 539.13 | 52.26 |
| Wavelength (Å) | 1.075 | 1.075 | 1.1 |
| Resolution (Å) | 39.56–4.3 (4.45–4.3)† | 49.61–4.40 (4.56–4.40)† | 49.32–2.40 (2.49–2.40)† |
| Total reflections | 1,204,932 (93,381) | 2,004,840 (73,134) | 138,950 (13,077) |
| Unique reflections | 153,939 (12,740) | 148,420 (10,172) | 22,705 (2257) |
| Multiplicity | 7.8 (6.2) | 13.5 (5.0) | 6.1 (5.8) |
| Completeness (%) | 99.6 (99.2) | 99.9 (99.6) | 100% (100%) |
| $<I>/\sigma I$ | 5.06 (0.65) | 9.10 (0.41) | 19.13 (1.66) |
| Wilson B-factor | 165.15 | 151.33 | 49.38 |
| $R_{pim}$‡ | 0.295 (1.61) | 0.138 (2.03) | 0.033 (0.44) |
| CC1/2§ | 0.948 (0.114) | 0.971 (0.166) | 0.998 (0.49) |
| CC*§ | 0.987 (0.453) | 0.993 (0.534) | 1.00 (0.811) |
| **Twinning** | | | |
| operator | – | – | –k, –h, –l |
| fraction | – | – | 0.42 |
| **Anisotropic scaling B-factors#** | | | |
| $a^*, b^*$ (Å$^2$) | 16.95 | 16.01 | – |
| $c^*$ (Å$^2$) | −33.90 | −32.03 | – |
| **Refinement** | | | |
| $R_{work}/R_{free}$ | 0.2748/0.3094 (0.3916/0.4100) | 0.2198/0.2639 (0.3660/0.3920) | 0.1629/0.1863 (0.2582/0.3036) |
| $CC_{work}/CC_{free}$§ | 0.928/0.890 (0.261/0.267) | 0.921/0.891 (0.318/0.262) | 0.870/0.498 (0.498/0.437) |
| No. atoms | 60,878 | 58,990 | 2753 |
| Protein/DNA | 60,872 | 58,984 | 2657 |
| Ligand/ion | 6 | 6 | 20 |
| Water | 0 | 0 | 76 |
| Protein residues | 7197 | 7195 | 342 |
| *B-factors* | | | |
| Protein | 179.52 | 194.66 | 60.35 |
| Ligand/ion | 158.99 | 139.48 | 49.77 |
| Water | – | – | 52.81 |
| R.m.s deviations | | | |
| Bond lengths (Å) | 0.005 | 0.004 | 0.010 |
| Bond angles (°) | 0.96 | 1.01 | 1.35 |

*Table 2. Continued on next page*

In the RPo structure, the $\sigma_2^A$ and $\sigma_3^A$ domains make extensive interactions with the promoter DNA (−17 to −4) from the (distorted) major groove side of the DNA, including critical interactions that maintain the upstream ds(−12)/ss(−11) junction of the transcription bubble (*Figure 1A–C*) (*Bae et al., 2015*). CarD does not make significant interactions with $\sigma^A$ but interacts with the promoter DNA from −14 to −10 from the opposite, (distorted) minor groove side of the DNA

Table 2. Continued

| | Holo-bubble-CarD | Holo-fork-CarD | CarD/β1-lobe |
|---|---|---|---|
| Clashscore | 19.58 | 14.83 | 19.72 |
| Ramachandran favored (%) | 88 | 89 | 91 |
| Ramachandran outliers (%) | 0.48 | 0.57 | 0.89 |

†Values in parentheses are for highest-resolution shell.
‡(*Diederichs and Karplus, 1997*).
§(*Karplus and Diederichs, 2012*).
#As determined by the UCLA MBI Diffraction Anisotropy Server (http://services.mbi.ucla.edu/anisoscale/).

(*Figure 1A–C*) such that the σ$^A$/DNA interactions and the structure of the transcription bubble in RPo and CarD/RPo are essentially the same (*Figure 1—figure supplement 6*). The KMnO$_4$ reactivity of thymine (T) bases within the transcription bubble (*Sasse-Dwight and Gralla, 1991*; *Ross and Gourse, 2009*) is identical in the presence or absence of CarD (*Figure 1D*, lanes 2 and 3), supporting the structural observation that the transcription bubble is the same with or without CarD. Although CarD does not alter the structure of the transcription bubble, it does increase the lifetime of RPo, as measured by the rate of disappearance of the KMnO$_4$ footprint after challenge with an excess of unlabeled competitor promoter (*Figure 1D*, lanes 4–7) (*Davis et al., 2015*).

The N-terminal ends of two CarD-CTD α-helices (α3 and α5) make direct contacts with the promoter DNA (*Figure 1C*, *Figure 2*). The two α-helices are positioned roughly perpendicular to the duplex DNA axis, forming a modest CarD/DNA interaction surface of 380 Å$^2$.

The peptide backbone nitrogen of CarD-L124, at the N-terminal end of α5, closely approaches the backbone phosphate oxygen of the template strand (t-strand) at the −14 position [−14(t)] (*Figure 2*), possibly forming a hydrogen bond, an interaction likely facilitated by the partial positive charge of the α5 helix dipole (*Hol et al., 1978*). Similar interactions have been observed in other DNA-binding proteins, such as helix-turn-helix proteins (*Harrison and Aggarwal, 1990*) and the nucleosome core particle (*Luger et al., 1997*).

## Role of a conserved CarD Trp residue in CarD function

W86 is conserved among greater than 95% of CarD proteins (*Figure 2—figure supplement 1*; *Source code 1*) and was shown to be important for CarD function as an activator (*Srivastava et al., 2013*). The bulky, hydrophobic planar side chain of W86, located at the N-terminal end of α3, wedges into the splayed minor groove at the upstream edge of the transcription bubble (*Figure 2*). Despite the relatively low resolution of our analysis (*Table 2*), CarD-W86 was clearly resolved in electron density maps (*Figure 2A*). The positioning of CarD-W86 was further supported by an unbiased simulated annealing omit $F_o − F_c$ map calculated from coordinates in which CarD-W86 had been mutated to Ala (*Figure 2C*).

Previous work showed that substitution of the bulky CarD-W86 side chain by Ala (*Tth* CarD-W86A or *Mtb* CarD-W85A) greatly reduced the activation efficiency of both *Tth* and *Mtb* CarD (*Srivastava et al., 2013*). To further evaluate the role of W86 in CarD function, we tested the activation efficiency of CarD harboring substitutions of W86 to other hydrophobic residues (A, F, Y, L, I and V) in an in vitro abortive transcription assay on the *Tth* 23S promoter (*Figure 3A*, *Figure 3—figure supplement 1*). All of the mutants tested showed impaired activity compared to wild-type CarD. A, F, and Y substitutions showed partial activation, while substitutions with branched-chain residues (I, L, V) showed no activation (I, V) or even a reduction of transcription compared to wild type CarD (L). Structural modeling suggests the branched-chain residues would clash with the DNA and interfere with CarD function.

The position of the W86 side chain Nε allows it to interact with T$_{−12}$(nt) O2 (*Figure 2B*). Since this mode of Trp/Thymine interaction is not common among DNA-binding proteins (*Lejeune et al., 2005*), we mutated the promoter −12 T/A base pair to C/G, G/C and A/T, and compared CarD activation at each of the three mutant promoters with its effect at the wild type promoter (we used *Mtb* AP3 for this analysis since *Tth* Eσ$^A$ was more active on this promoter than on *Tth* 23S, allowing us to analyze the weak activity of the mutant promoters). The −12 T/A base pair is a conserved part of the promoter −10 element (*Shultzaberger et al., 2007*), and, as expected, transcription activity from each of the mutant promoters was reduced at least threefold (*Figure 3B*, *Figure 3—figure supplement 1*) (*Moyle et al., 1991*). In addition, CarD activation on each mutated promoter was substantially lower than on the wild

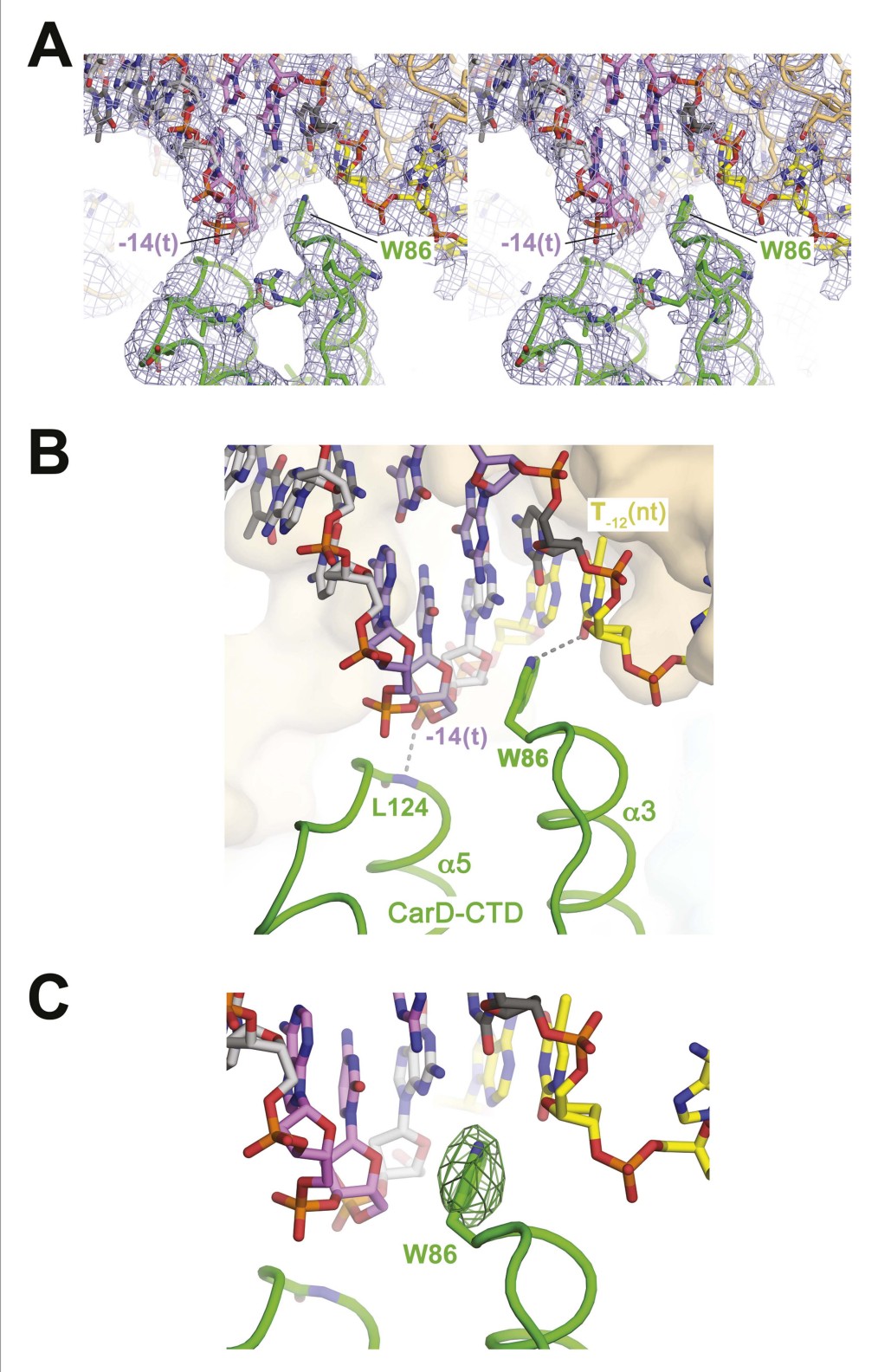

**Figure 2**. CarD-CTD/promoter DNA interactions. (**A**) Stereo view of the refined, B-factor sharpened (−80 Å²) $2F_o − F_c$ map (grey mesh, contoured at 1σ), with superimposed DNA and CarD. Density for the close approach of the CarD peptide backbone to the −14(t) DNA phosphate backbone and for CarD-W86 are clearly resolved. (**B**) Close up view showing interactions between the N-terminal ends of α3 and α5 of the CarD-CTD with promoter DNA at the upstream

*Figure 2. Continued*

ds(−12)/ss(−11) junction of the transcription bubble. Grey dashed lines indicate potential polar interactions between the peptide backbone nitrogen of L124 and the −14(t) phosphate oxygen, and W86 Nε and O2 of $T_{−12}$(nt). (**C**) Same view as *Figure 2B*. Superimposed is the simulated annealing omit map (dark green mesh, $F_o − F_c$, contoured at 3σ), calculated from a model where CarD-W86 was mutated to Ala. The unbiased difference Fourier density shows that the side chain position is specified in the data.

The following figure supplement is available for figure 2:

**Figure supplement 1**. Alignment of CarD homologs found in bacteria from 11 diverse phyla/groups.

type promoter (*Figure 3B*), suggesting that the observed interaction between W86 and $T_{−12}$(nt) contributes to CarD activity.

Note that the CarD W86F substitution results in an approximately twofold loss in CarD fold activation (*Figure 3A*, threefold activation for wild-type CarD vs 1.5-fold for W86F), as does substitution of the promoter −12 bp by anything other than the wild-type T/A bp (*Figure 3B*). The Phe side chain at CarD position 86 would be expected to fulfill the stacking and steric roles of CarD-W86 effectively, but would not be able to participate in the putative H-bond with the $T_{−12}$(nt) O2 atom. We tentatively suggest that the reduced activation efficiency of the CarD-W86F mutant is primarily due to the loss of the minor groove polar interaction with $T_{−12}$(nt).

## The *Thermus* CarD/RNAP initiation complex structures represent the active conformation of CarD

A crystal structure of *Mtb* CarD in complex with an *Mtb* RNAP β-subunit fragment that includes the β1-lobe shows a relative orientation of the CarD-RID/CarD-CTD domains very different from the one in our *Tth* CarD structures, despite high sequence and structural similarity within the domains (*Gulten and Sacchettini, 2013*; *Srivastava et al., 2013*). In the *Mtb* structure, the CarD-CTD is rotated ~140° relative to the CarD-RID (*Figure 4A*). Structural modeling in the context of RPo positions the *Mtb* CarD-CTD and the functionally important W85 away from the promoter DNA (*Figure 4A,B*). To determine the functional conformation of CarD, we introduced a disulfide to lock the conformation of *Mtb* CarD into the one observed in the *Tth* CarD structures. In the seven crystallographically independent copies of *Tth* CarD (PDB IDs 4L5G and structures reported here) (*Srivastava et al., 2013*), the average distance between the α-carbons of CarD-RID-P13 and CarD-CTD-G100 is 5.7 ± 0.8 Å, and among the four copies determined in the presence of promoter DNA, an even tighter distribution is observed, 5.2 ± 0.1 Å (*Figure 4B*, right). On the other hand, the corresponding positions in the *Mtb* CarD structure (P12/G99) are 24 Å apart (*Figure 4B*, left) (*Gulten and Sacchettini, 2013*). Cys substitutions at these positions are predicted to form a disulfide bond under oxidizing conditions in the *Tth* CarD conformation (thus locking the domain orientation), but not the *Mtb* CarD conformation (*Figure 4B*). We engineered the P12/G99 Cys substitutions in *Mtb* CarD (*Mtb* CarD2C; wild-type *Mtb* CarD is devoid of Cys residues). Non-reducing SDS polyacrylamide gel electrophoresis and liquid chromatography mass spectrometry confirmed that under oxidizing conditions, the CarD-RID and CarD-CTD were disulfide crosslinked in greater than 98% of CarD2C, while under reducing conditions, no disulfide bond was present in >99% of CarD2C (*Figure 4C*). We tested the function of oxidized (crosslinked) and reduced CarD2C using a mycobacterial transcription system (*Srivastava et al., 2013*; *Davis et al., 2015*) on the *Mtb* AP3 promoter. Under oxidizing conditions, the cross-linked, conformationally locked CarD2C activated transcription as well as wild type CarD (*Figure 4D*, 0 mM dithiothreitol [DTT], *Figure 4—figure supplement 1*). The observation that under reducing conditions, CarD2C was somewhat impaired in transcription activation (*Figures 4D, 5* mM DTT) is explained by the fact that the CarD positions corresponding to *Mtb* CarD P13 and G99 are conserved (*Srivastava et al., 2013*; *Figure 2—figure supplement 1*, *Source code 1*); on this basis substitution of these positions would be expected to impair uncrosslinked CarD2C function. We conclude that the *Tth* CarD structures, with CarD-CTD W86 positioned to interact with the upstream edge of the transcription bubble (*Figure 2*), represents the functional conformation of CarD.

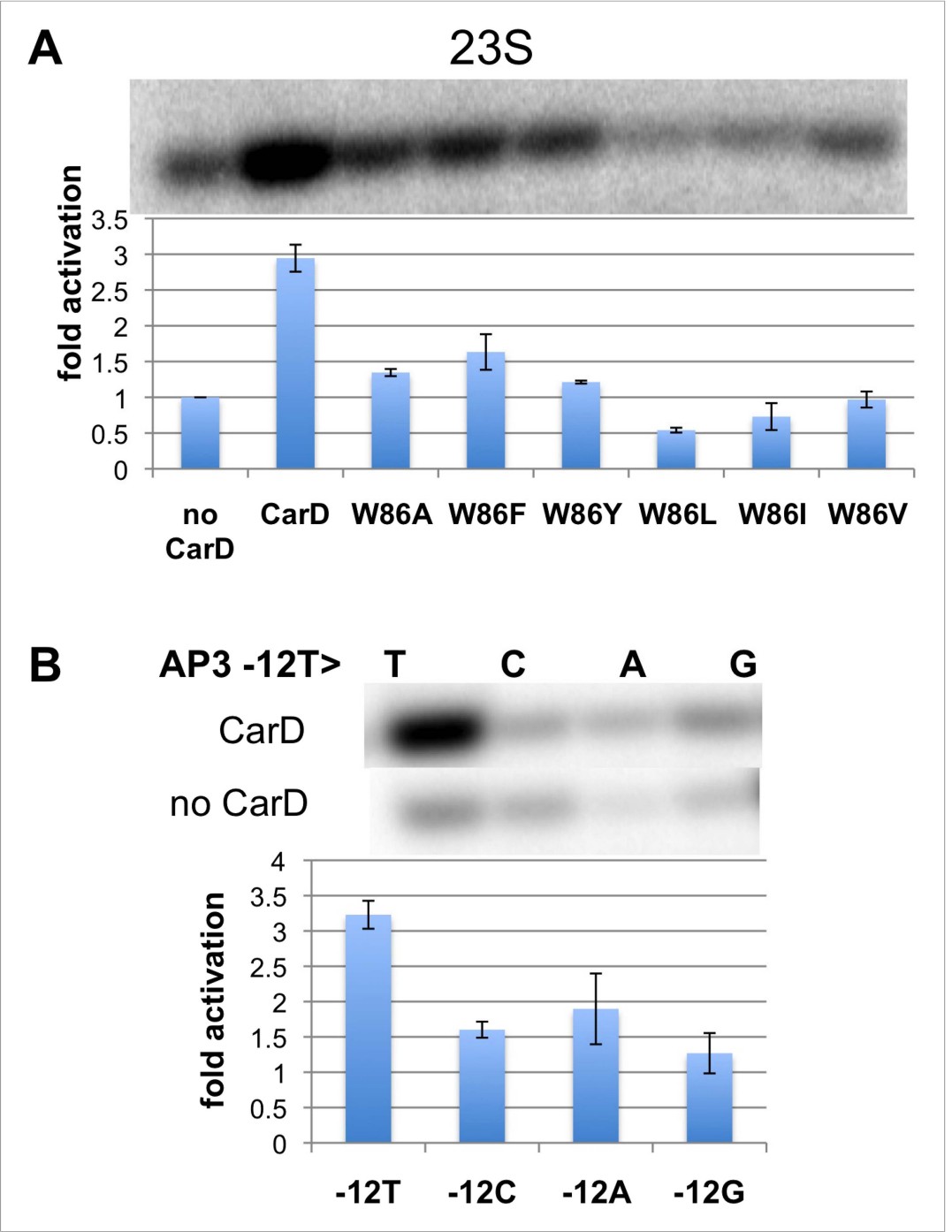

**Figure 3**. Function of CarD-W86. (**A**) The effect of CarD-W86 substitutions on activation of abortive initiation (UpG dinucleotide + α-$^{32}$P-CTP) on the *Tth rrnA-23S* promoter (normalized with respect to no CarD). Error bars denote the standard error from a minimum of three experiments. (**B**) The effect of promoter −12 base pair substitutions on activation of abortive initiation (GpU dinucleotide + α-$^{32}$P-UTP) by CarD on the *Mtb rrnA*-AP3 promoter. Error bars denote standard errors.

The following figure supplement is available for figure 3:

**Figure supplement 1**. Complete gels for the abortive initiation assays shown in (**A**) *Figure 3A* and (**B**) *Figure 3B*.

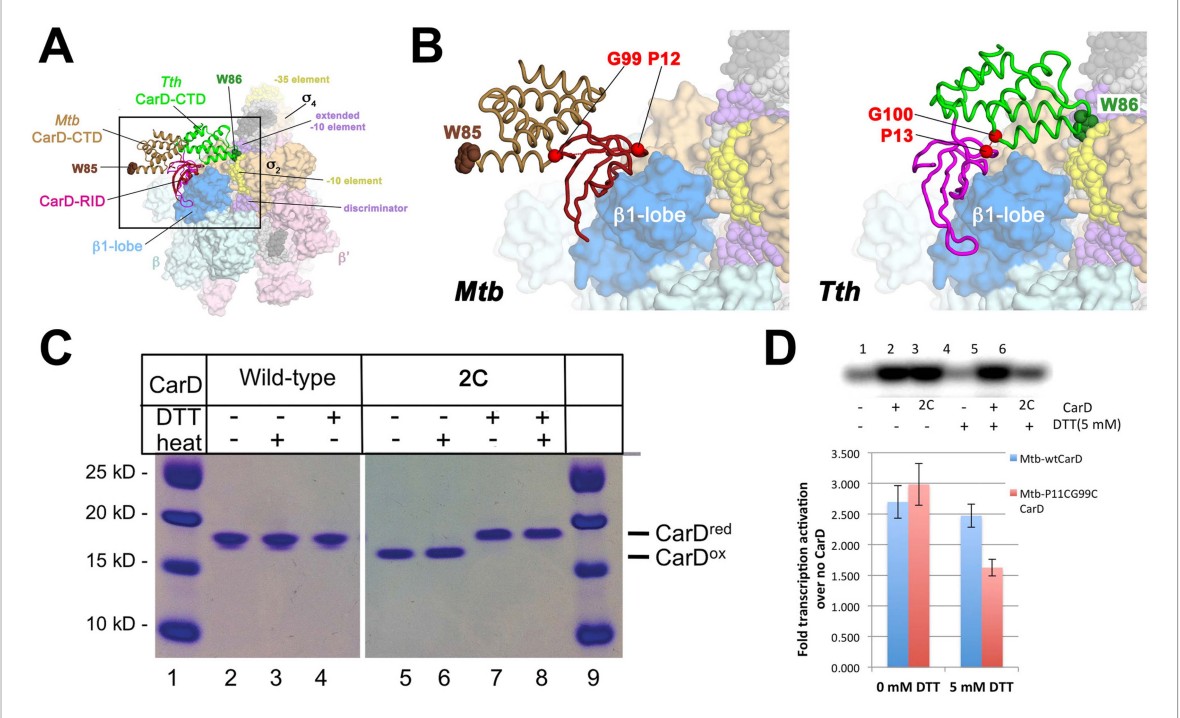

**Figure 4**. Inter-domain crosslinking confirms the functional conformation of CarD. (**A**) View of the Thermus CarD/RPo complex. RNAP holoenzyme and nucleic acids are shown as in *Figure 1B*; *Tth* CarD is shown as an α-carbon ribbon (*Tth* CarD-RID, magenta; *Tth* CarD-CTD, green) but with W86 shown in CPK format and colored dark green. Also shown is *Mtb* CarD from the *Mtb* CarD/β1-β2-lobe structure (4KBM; *Mtb* CarD-RID, dark red; *Mtb* CarD-CTD, brown, but with W85 colored dark brown), superimposed by alignment of 145 Cα atoms from the β1-lobe (1.39 Å rmsd). The boxed region is magnified in (**B**). (**B**) (*Left*) Magnified view showing the modeled *Mtb* CarD in the context of RPo. The α-carbons of CarD-RID-P12 and CarD-CTD-G99, shown as red spheres, are ~24 Å apart (red dashed line). A disulfide bond between these two positions in *Mtb* CarD2C (P12C/G99C substitutions) would disallow this conformation of CarD. (*Right*) Magnified view of the Thermus CarD/RPo complex. CarD-RID-P13 and CarD-CTD-G100 are ~5.2 Å apart (red dashed line). A disulfide bond between the corresponding two positions in *Mtb* CarD2C would lock this DNA-interacting conformation of CarD. (**C**) Purification of disulfide crosslinked (lanes 5, 6) and reduced (lanes 7, 8) CarD2C. Non-reducing SDS-PAGE illustrates that CarD2C is oxidized (crosslinked) in the absence of reducing agent dithiothreitol (DTT) and is reduced (uncrosslinked) in the presence of DTT. Samples were excised from gels and LC-MS was used to confirm oxidation states. (**D**) Effect of oxidation state on *Mtb* CarD2C activation of abortive transcription on the *Mtb* AP3 promoter (GpU dinucleotide + α-$^{32}$P-UTP). Conformationally locked (no DTT) *Mtb* CarD2C exhibits wild type activation activity.

The following figure supplement is available for figure 4:

**Figure supplement 1**. Complete gel for the abortive initiation assay shown in *Figure 4D*.

## CarD stabilizes RPo by preventing transcription bubble collapse

CarD may stabilize RPo by forming favorable interactions with the upstream edge of the unwound transcription bubble (*Figures 1C, 2B*). We tested the lifetime of competitor-resistant RPo challenged with a competitive promoter trap (*Davis et al., 2015*) using the abortive initiation assay on both the 23S and AP3 promoters (*Figure 5A*). *Tth* CarD increased the half-life ($t_{1/2}$) of the *Thermus* RPo ~threefold on each promoter (*Figure 5B*, *Figure 5C*, *Figure 5—figure supplement 1*, *Figure 5—figure supplement 2*) while *Eco* RNAP did not dissociate significantly from either promoter over the lifetime of the experiments ($t_{1/2} >> 120$ min; *Figure 5B*) (*Davis et al., 2015*). The *Tth* CarD W86A substitution diminished or abolished the ability of CarD to increase $t_{1/2}$ on the 23S and AP3 promoter, respectively (*Figure 5B,C*).

Clearly, dissociation of RPo and transcription bubble collapse (rewinding) are closely linked. We hypothesized that CarD may increase the lifetime of RPo by preventing transcription bubble collapse. To test this hypothesis, we determined the effect of CarD on the lifetime of promoter complexes on a synthetic promoter template based on the 23S sequence and compared it with the same synthetic template but with a non-complementary transcription bubble (from −11 to +2) unable

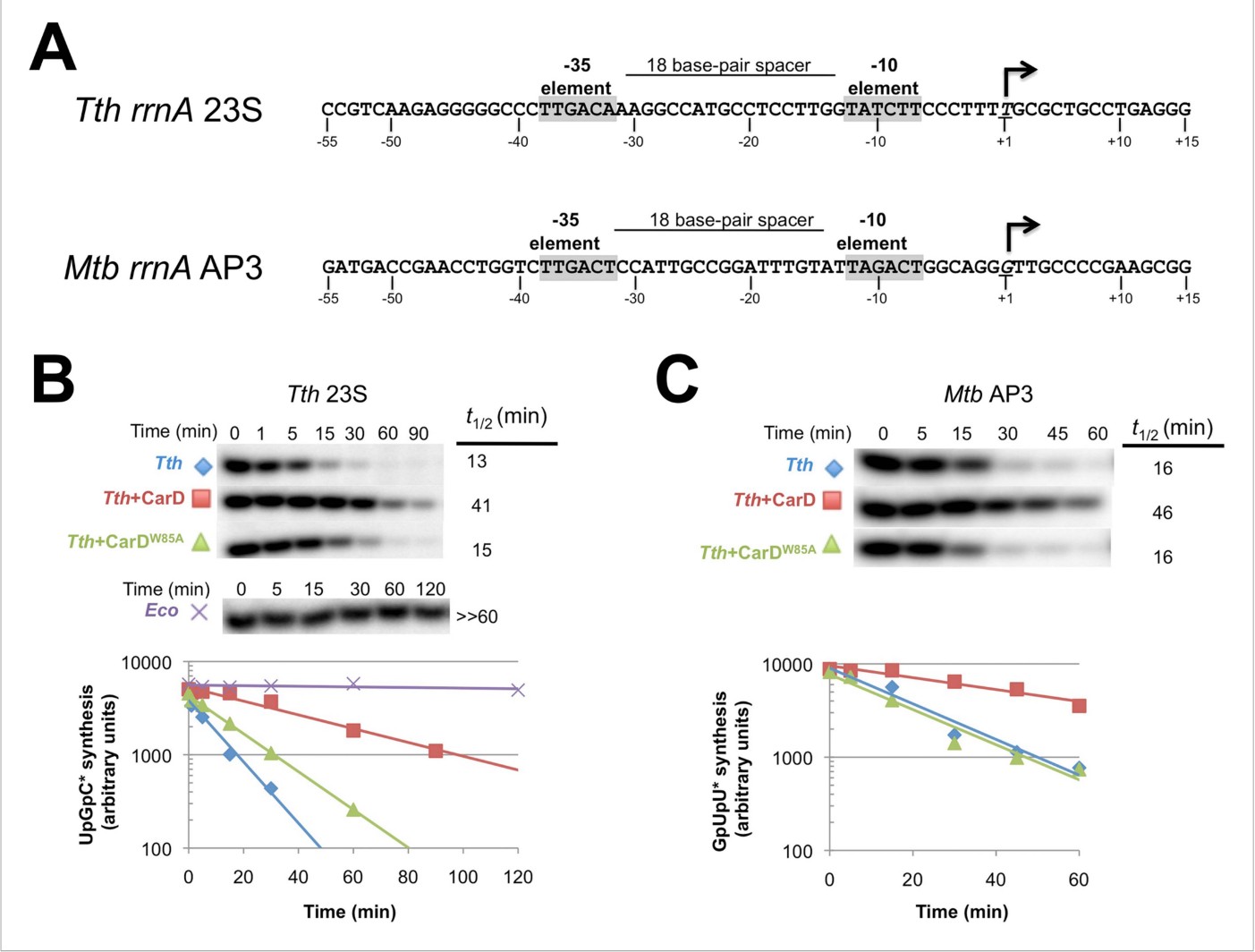

**Figure 5**. CarD increases the lifetime of Thermus RPo. (**A**) Sequences of *Mtb rrnA*AP3 (*Gonzalez-y-Merchand et al., 1996*) and *Tth* 23S rRNA (*Hartmann et al., 1987*) promoters used in in vitro assays. (**B**, **C**) Lifetimes of promoter complexes measured by abortive transcription. At the top of each panel, [$^{32}$P]-labeled abortive transcript production at times after addition of a large excess of competitor promoter DNA trap was monitored by polyacrylamide gel electrophoresis and autoradiography. On the bottom, transcript production was quantified by phosphorimagery and plotted. The lines indicate single-exponential decay curves fit to the data points. The calculated decay half-lives ($t_{1/2}$) are shown to the right of the gel images. Assays were performed on the following templates: (**B**) *Tth rrnA-23S* promoter (UpG dinucleotide + α-$^{32}$P-CTP). (**C**) *Mtb rrnA*-AP3 promoter (GpU dinucleotide + α-$^{32}$P-UTP).

The following figure supplements are available for figure 5:

**Figure supplement 1**. Complete gels for the abortive initiation assays shown in *Figure 5B*.

**Figure supplement 2**. Complete gels for the abortive initiation assays shown in *Figure 5C*.

to collapse (*Figure 6A*). On the duplex template (23S_DS), CarD increased the $t_{1/2}$ more than fivefold (*Figure 6B*, *Figure 6—figure supplement 1*). On the bubble template (*Figure 6A*, 23S_Bub) in the absence of CarD, the $t_{1/2}$ was also increased more than fivefold, indicating that the relatively short lifetime of *Tth* RPo on the 23S promoter is due largely to bubble collapse (*Figure 6B*). Addition of CarD to the bubble template had no effect on the level of transcription and did not affect RPo lifetime (*Figure 6B*). We thus conclude that a primary function of *Tth* CarD, like *Mtb* CarD (*Davis et al., 2015*), is to stabilize RPo by preventing collapse of the transcription bubble.

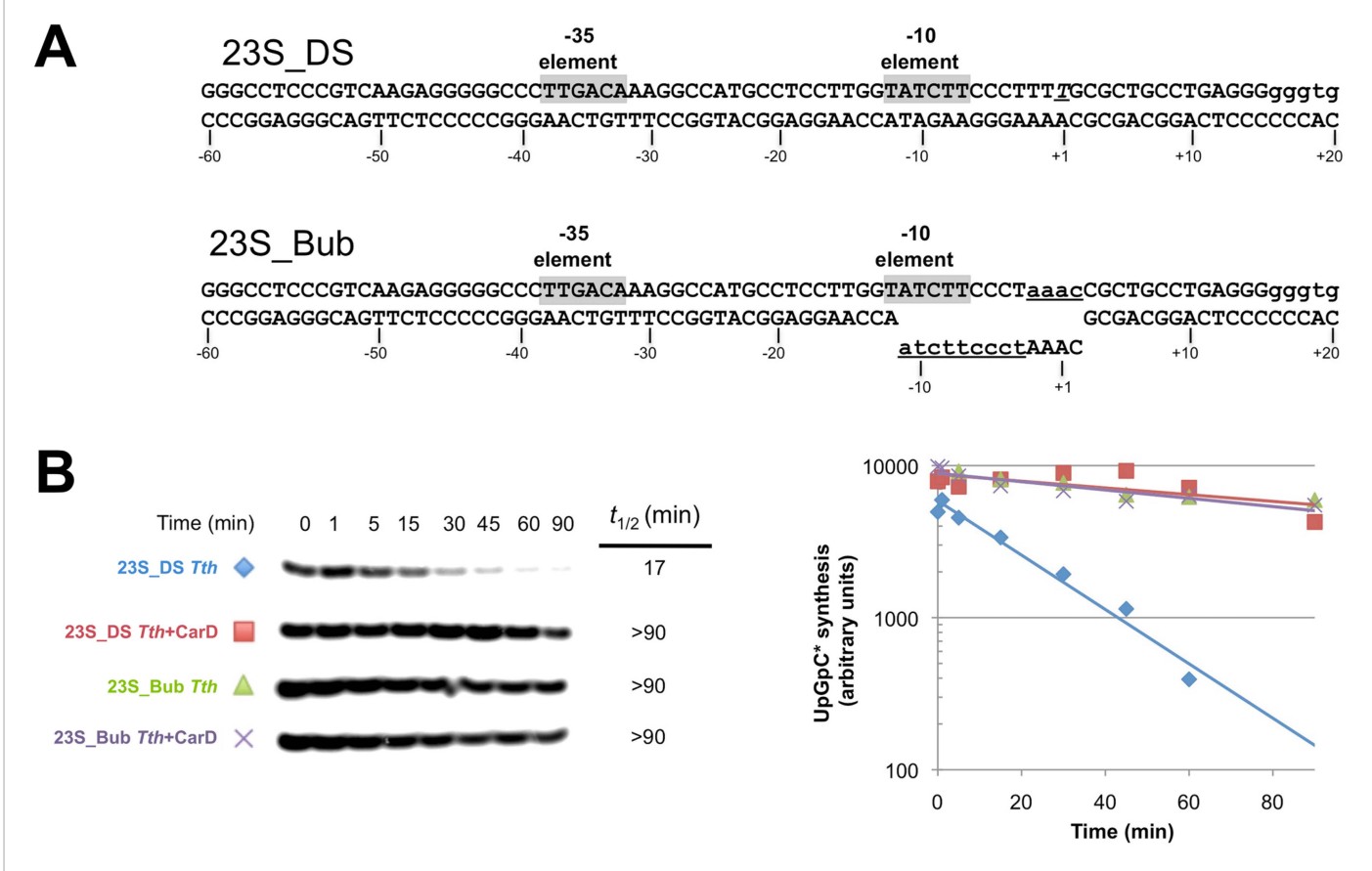

**Figure 6**. CarD increases the lifetime of Thermus RPo by preventing transcription bubble collapse. (**A**) Synthetic duplex (23S_DS) and artificial bubble (23S_Bub) promoters used in in vitro assays. (**B**) Lifetimes of promoter complexes formed on synthetic templates measured by abortive transcription (UpG dinucleotide + $\alpha$-$^{32}$P-UTP). (*Left*) [$^{32}$P]-labeled abortive transcript production at times after addition of a large excess of competitor promoter DNA trap was monitored by polyacrylamide gel electrophoresis and autoradiography. (*Right*) transcript production was quantified by phosphorimagery and plotted. The lines indicate single-exponential decay curves fit to the data points. The calculated decay half-lives ($t_{1/2}$) are shown to the right of the gel images. Assays were performed on the synthetic double-stranded (23S_DS) and bubble (23S_Bub) templates.

The following figure supplement is available for figure 6:

**Figure supplement 1**. Complete gels for the abortive initiation assays shown in *Figure 6B*.

## Discussion

CarD is an essential transcription activator in *Mtb* that is also widely distributed among bacterial species, including *Thermus* species but not found in *Eco* (*Stallings et al., 2009*; *Srivastava et al., 2013*; *Table 1*). In the absence of a structure of a mycobacterial transcription initiation complex, we present here the structure of *Tth* CarD with a *Taq* transcription initiaton complex (*Figure 1B*). The structural results, combined with supporting biochemical studies, establish that the CarD-RID makes a protein/protein interaction with the RNAP β1-lobe, thereby positioning the CarD-CTD and a conserved Trp residue to interact with the upstream edge of the transcription bubble, using a wedge mechanism to prevent collapse of the transcription bubble (*Figures 1C, 2B*). This is a previously unseen mechanism of activation by a transcription factor. Specifically: (1) CarD does not induce any major changes on the holoenzyme nor the transcription bubble (*Figure 1D*); (2) CarD contacts with the DNA are mostly confined to the backbone phosphates, with the exception of the conserved Trp (W86) that serves as a wedge at the upstream edge of the bubble, which may be stabilized by a hydrogen bond with the conserved T$_{-12}$(t) (*Figure 2B*). We show that this W-wedge residue and its interaction with T$_{-12}$(t) is important for full CarD function (*Figure 3*); (3) We show that *Tth* CarD functions similarly

to *Mtb* CarD to increase the lifetime of RPo by preventing collapse of the transcription bubble (*Figure 5*, *Figure 6*); (4) We show that the mode of CarD interaction with RNAP and with the promoter DNA revealed by our structures represents the functionally relevant conformation (*Figure 4*), resolving conflicting models.

*Eco* has served as a model organism for the study of many cellular processes over the last few decades, including transcription. *Eco* RNAP forms unusually stable RPo and *Eco* lacks CarD, while RNAPs shown to form relatively unstable RPo come from bacteria that harbor CarD (*Bacillus subtilis*, *Whipple and Sonenshein, 1992*; *Artsimovitch et al., 2000*; *Mtb*, *Davis et al., 2015*; *Myxococcus xanthus*, *Gallego-Garcia et al., 2014*; *Taq*, *Miropolskaya et al., 2012*; *Tth*, *Xue et al., 2000*). Moreover, CarD is a global regulator, found at most $\sigma^A$ promoters throughout the *Mycobacterium smegmatis* genome (*Srivastava et al., 2013*), and is essential in the two mycobacterial species where it has been tested (*Stallings et al., 2009*). These observations suggest that CarD boosts transcription at most (if not all) promoters by acting as a basal transcription factor, required to compensate for otherwise rapidly dissociating RNAP/promoter complexes. While the interaction with $T_{-12}(t)$ (*Figure 2B*, *Figure 3B*) may modulate the effect of CarD in a promoter-specific manner, this is likely to be a minor effect in vivo since $T_{-12}(t)$ is present at most $\sigma^A$ promoters (*Shultzaberger et al., 2007*), and CarD can nevertheless activate transcription from promoters that lack a −12 T/A base pair (although less effectively; *Figure 3B*).

The structural and biochemical studies of *Tth* CarD/RPo complexes presented here reveal how the widely distributed transcription factor CarD interacts with RPo to activate initiation. The CarD-RID/RNAP β1-lobe protein/protein interaction positions the CarD-CTD to interact with the upstream edge of the transcription bubble in a functionally relevant pose that does not allosterically alter the structure of the transcription bubble nor RNAP holoenzyme/promoter interactions (*Figure 1D*, *Figure 1—figure supplement 6*); instead CarD supports pre-existing RNAP holoenzyme/promoter DNA interactions in RPo. The mode of CarD/DNA interaction is incompatible with duplex B-form DNA; the normal minor groove is too narrow to accommodate the end of CarD-α3 and CarD-W86 (*Srivastava et al., 2013*). This is consistent with a kinetic analysis of CarD function that concluded CarD stabilizes RPo by increasing the rate of isomerization to RPo and decreasing the rate of bubble collapse, but has little effect on the formation of the closed RNAP/promoter complex (*Rammohan et al., 2015*).

The CarD contacts with the DNA occur mostly through the backbone phosphates, except for highly conserved CarD-W86, which wedges between the splayed DNA strands at the upstream edge of the transcription bubble and may form a hydrogen bond with $T_{-12}(t)$ O2 presented in the highly distorted minor groove (*Figure 2B*). The minor groove W-wedge increases the lifetime of RPo by preventing transcription bubble collapse (*Figure 6*). While we believe this is the dominant mode of action for CarD, CarD may affect other steps of the initiation process as well. This previously unseen mode of transcription activation may be absent in *Eco* (the focus of most mechanistic transcription studies) since *Eco* RNAP forms relatively stable complexes on most promoters (*Figure 5B*) (*Davis et al., 2015*).

## Materials and methods

### Crystallization of *Thermus* CarD/RNAP holoenzyme/promoter complexes

Crystals of *Taq* Δ1.1σ^A-holoenzyme/promoter complexes were grown as described (*Bae et al., 2015*). *Tth* CarD (prepared as described previously; *Srivastava et al., 2013*), in 1 mM in 20 mM Tris-HCl, pH 8.0, 0.2 M NaCl, was added directly to the hanging drops containing RPo crystals to a final concentration of 100 μM. After 1 day of incubation, the crystals were cryo-protected and frozen as described (*Bae et al., 2015*).

### Structure determination of *Thermus* CarD/RNAP holoenzyme/promoter complexes

X-ray diffraction data were collected at Brookhaven National Laboratory National Synchrotron Light Source (NSLS) beamline X29. Data were integrated and scaled using HKL2000 (*Otwinowski and Minor, 1997*). The diffraction data were anisotropic. To compensate, isotropy was approximated by applying a positive b factor along a* and b* and a negative b factor along c* (*Table 1*), as implemented by the UCLA MBI Diffraction Anisotropy Server (http://services.mbi.ucla.edu/anisoscale/) (*Strong et al., 2006*), resulting in enhanced map features (*Figure 2A,C*).

Initial Fourier difference maps, calculated after rigid body refinement (*Adams et al., 2010*) starting with the appropriate RNAP-holoenzyme/promoter complex structure (*Bae et al., 2015*), revealed clear density corresponding to CarD. CarD was docked into the maps with the aid of a 2.4 Å-resolution structure of a *Tth* CarD/*Taq* β1-lobe complex (PDB ID 4XAX, *Figure 1—figure supplement 3*, *Table 2*, see below). The models were improved in further steps of refinement: (1) rigid body refinement of 20 individual mobile domains in RNAP and 2 domains of CarD (CarD-RID and CarD-CTD) (*Adams et al., 2010*); (2) deformable elastic network refinement (*Schröder et al., 2010*) with noncrystallographic symmetry restraints using CNS 1.3 (*Brunger et al., 1998*) performed on the Structural Biology Grid portal (*O'Donovan et al., 2012*); (3) iterative cycles of manual building with COOT (*Emsley and Cowtan, 2004*) and refinement with PHENIX (*Adams et al., 2010*). The PDBePISA server (http://www.ebi.ac.uk/pdbe/pisa/) was used to calculate intermolecular buried surface areas (*Krissinel and Henrick, 2007*).

## Resolution limit and structure validation

We follow the criteria of *Karplus and Diederichs (2012)*, as explained in the accompanying paper (*Bae et al., 2015*).

In the final $2F_o - F_c$ electron density maps, the CarD-W86 side chain was clearly resolved (*Figure 2A*). To confirm the side chain position, we produced an unbiased difference Fourier map using a simulated annealing omit procedure. The CarD-W86 side chain was removed from the structural model by mutation to Ala, and the mutated models were subjected to simulated annealing refinement (2500 K) using PHENIX (*Adams et al., 2010*) (*Figure 2C*).

## Crystallization of *Thermus* CarD/β1-lobe complex

The plasmids pET21a *Taq*β1 (*Westblade et al., 2010*) and pETsumo*Tth*CarD (*Srivastava et al., 2013*) were separately transformed into *Eco* BL21(DE3) cells (EMD Millipore, Billerica, MA, United States) and transformants were grown at 37°C in Luria–Bertani media containing ampicillin (200 μg/ml) and kanamycin (50 μg/ml). At an $A_{600nm}$ between 0.6–0.8, the cultures were supplemented with isopropyl-β,D-thiogalactopyranoside (0.5 mM final concentration) to induce protein expression for 4 hr at 30°C. The cells were then spun down by centrifugation and resuspended in buffer A (20 mM Tris-HCl, pH 8.0 at 4°C, 500 mM NaCl, 5 mM imidazole, 5% (vol/vol) glycerol, 1 mM β-mercaptoethanol). The cells were lysed using a continuous-flow homogenizer (Avestin Inc., Ottawa, ON, Canada) and then centrifuged to remove insoluble debris. The clarified cell lysate containing overexpressed (His)$_{10}$Sumo-*Tth* CarD was first applied to a Ni$^{2+}$-charged HiTrap column (GE Healthcare Bio-Sciences, Marlborough, MA, United States) that was equilibrated with buffer A, followed by a wash with five column volumes (cv) of buffer A + 25 mM imidazole. Subsequently, clarified cell lysate containing overexpressed *Taq* β1 was injected into the same column to form a complex with the immobilized (His)$_{10}$Sumo *Tth* CarD. The column was washed with five column volumes (cv) of buffer A + 25 mM imidazole and 5 cv buffer A + 40 mM imidazole. The complex bound to the column was eluted with buffer A + 250 mM imidazole. After overnight cleavage with UlpI protease (GE Healthcare) to remove the (His)$_{10}$Sumo-tag from *Tth* CarD and dialysis against buffer A + 25 mM imidazole, a subtractive Ni$^{2+}$-chelating chromatographic step removed uncleaved (His)$_{10}$Sumo-*Tth* CarD and the cleaved (His)$_{10}$Sumo-tag. The sample was concentrated and injected on a Superdex 75 gel filtration column (GE Healthcare) that was equilibrated with GF buffer (50 mM MES-OH, pH 6.5, 500 mM NaCl, 5% (vol/vol) glycerol). Fractions containing purified *Tth* CarD/*Taq* β1 complex were pooled and concentrated to 15 mg/ml by centrifugal filtration. Sodium dodecyl sulfate polyacrylamide gel electrophoresis and Coomassie blue staining were used to analyze the purity of the complex.

Crystals were grown by hanging-drop vapor diffusion by mixing 1 μl of protein solution (15 mg/ml in GF buffer) with 1 μl of crystallization solution (1.5 M ammonium sulfate, 0.1 M sodium acetate, pH 5.0, 25% (vol/vol) ethylene glycol) and incubating over a well containing crystallization solution at 22°C. Large crystals (0.5 mm) grew within 1 day. The crystals were directly frozen in liquid nitrogen for data collection.

## Structure determination of the *Thermus* CarD/β1-lobe complex

X-ray diffraction data were collected at Brookhaven National Laboratory NSLS beamline X29. Data were integrated and scaled using HKL2000 (*Otwinowski and Minor, 1997*) (*Table 1*).

Initial electron density maps were calculated by molecular replacement using Phaser (*McCoy et al., 2007*) from starting models of the *Taq* β1-lobe (2.9 Å-resolution; 3MLQ; *Westblade et al., 2010*) and *Tth* CarD (2.4 Å-resolution; 4L5G; *Srivastava et al., 2013*). One CarD/β1-lobe complex was clearly identified in the asymmetric unit. The model was first adjusted manually using COOT (*Emsley and Cowtan, 2004*), then further refined using the Autobuild feature of PHENIX (*Adams et al., 2010*). At this point, the model fit well to the electron density but the $R_{free}$ and $R$ factors remained relatively high (>0.3). Twinning was identified by Xtriage in PHENIX (twinning operators −k, −h, −l; twinning fraction 0.42). The final model was obtained after twinning refinement using PHENIX.

## Promoter DNA used in biochemical assays

To prepare the promoter DNAs, fragment −86 to +70 of pUC57-*MtbrrnA*P3 (−60 to +15 of the endogenous promoter sequence) was prepared as described (*Davis et al., 2015*). Fragment −171 to +69 of pRLG6768-*Tthrrn23S* (−68 to +15 of the endogenous promoter sequence) (*Vrentas et al., 2008*) was prepared similarly to AP3. These fragments (AP3 and 23S) served as templates for all transcription assays unless otherwise noted. AP3 −12T substitutions were synthesized (GenScript, Piscataway, NJ, United States) and placed into pUC57 and prepared as described for AP3 (*Davis et al., 2015*). Artificial bubble and double-stranded templates of 23S (−60 to +20) were synthesized as oligonucleotides and gel purified (IDT; *Figure 6A*). The purified oligonucleotides were annealed and used as templates for assays.

## KMnO$_4$ footprinting

KMnO$_4$ footprinting on the *Mtb rrnA*P3 promoter (*Figure 1D*) was performed as described (*Davis et al., 2015*) except reactions were at 65°C with 100 mM NaCl.

## Transcription assays

Abortive initiation assays (*Figure 3*, *Figure 3—figure supplement 1*, *Figure 4D*, *Figure 4—figure supplement 1*, *Figure 5B*, *Figure 5C*, *Figure 5—figure supplement 1*, *Figure 5—figure supplement 2*, *Figure 6B*, *Figure 6—figure supplement 1*) were performed as previously described (*Srivastava et al., 2013*; *Davis et al., 2015*) with the following adaptations for the *Thermus* transcription system. Briefly, reactions were performed in transcription buffer (10 mM Tris-HCl, pH 8.0, 1 mM MgCl$_2$, 0.1 mM DTT, 50 µg/ml BSA) with 100 mM NaCl for the AP3 promoter or 100 mM K-glutamate for the 23S promoter, at 65°C. Core RNAP (200 nM) and σ$^A$ (1 µM) were combined and incubated at 65°C for 5 min to form holoenzyme. CarD (2 µM, when used) was then added to the holoenzyme and incubated for an additional 5 min. Next, promoter DNA (10 nM) was added and RPo was allowed to form for 15 min at 65°C. Abortive transcription was initiated by the addition of an NTP mix containing the initiating dinucleotide (250 µM, GpU for AP3, UpG for 23S; TriLINK), the next NTP (α-$^{32}$P-labeled, UTP for AP3, CTP for 23S; 1.25 µCi, with 50 µM of the same unlabeled NTP) and 2 µM of FC-bubble competitor DNA when used (*Figure 1D*) (*Davis et al., 2015*). After 10 min, transcription was quenched and analyzed as previously described (*Davis et al., 2015*). For half-life assays, competitor was first added and NTP substrates were added at different times as indicated (*Figures 5B,C, 6B*).

## *Mtb* CarD substitution mutants

Single amino acid substitutions of CarD W86 were generated using site-directed mutagenesis (Stratagene-Agilent Technologies, Santa Clara, CA, United States) and purified using the same procedure as wild-type CarD (*Srivastava et al., 2013*). *Mtb* CarD2C (P12C/G99C) was also made using site-directed mutagenesis but was subjected to two additional purification steps. Tandem Q-sepharose column chromatography (GE Healthcare) was used to remove inter-molecular cross-linked CarD. Sample was first applied on a 5 ml column and eluted using a NaCl gradient from 200 mM to 1 M over 20 column volumes (cv). The purest fractions were combined and reapplied to a second 5 ml Q column and eluted using a NaCl gradient from 100 mM to 1 M over 40 cv. This purification yielded >95% intra-molecular cross-linked CarD as verified by non-reducing SDS-PAGE (*Figure 3C*) and liquid chromatography-mass spectrometry-MS analysis (The Rockefeller University Proteomics Resource Center). Transcription assays with *Mtb* CarD were performed similarly to the *Thermus* assays in the same transcription buffer but at 37°C with 10 mM K-Glutamate rather than 100 mM NaCl. Transcription at reducing conditions included 5 mM DTT, at oxidizing conditions no DTT was present.

## Accession numbers

The structure factor files and X-ray crystallographic coordinates have been deposited in the Protein Data Bank under ID codes 4XLS (*Tth* CarD/*Taq* holoenzyme/us-fork (− 12 bp) complex), 4XLR (*Tth* CarD/*Taq* RPo), and 4XAX (*Tth* CarD/*Taq* β1-lobe).

## Acknowledgements

We thank D Oren and The Rockefeller University Structural Biology Resource Center for technical assistance (supported by grant number 1S10RR027037 from the National Center for Research Resources of the NIH). LC-MS to confirm disulfide cross-linking was performed at The Rockefeller University Proteomics Resource Center by J Fernandez. We thank R Saecker for helpful discussions and R Gourse and W Ross for plasmids harboring the *Tth rrnA-23S* promoter. We thank W Shi (NSLS beamline X29) for support with synchrotron data collection. This work was based, in part, on research conducted at the NSLS and supported by the U.S. Department of Energy, Office of Basic Energy Sciences and by the Center for Synchrotron Biosciences grant, P30-EB-009998, from the National Institute of Biomedical Imaging and Bioengineering (NIBIB). BB was supported by a Merck Postdoctoral Fellowship (The Rockefeller University) and an NRSA (NIH F32 GM103170). This work was supported by R01 GM114450 to EAC.

## Additional information

### Funding

| Funder | Grant reference | Author |
|---|---|---|
| National Center for Research Resources (NCRR) | 1S10RR027037 | Seth A Darst |
| National Institute of Biomedical Imaging and Bioengineering (NIBIB) | P30-EB-009998 | Seth A Darst |
| The Rockefeller University | Merck Postdoctoral Fellowship | Brian Bae |
| National Institute of General Medical Sciences (NIGMS) | F32 GM103170 | Brian Bae, Elizabeth A Campbell |

The funders had no role in study design, data collection and interpretation, or the decision to submit the work for publication.

### Author contributions

BB, JC, Conception and design, Acquisition of data, Analysis and interpretation of data; ED, KL, Acquisition of data, Analysis and interpretation of data; SAD, Conception and design, Acquisition of data, Analysis and interpretation of data, Drafting or revising the article; EAC, Conception and design, Analysis and interpretation of data, Drafting or revising the article

## Additional files

### Supplementary file

• Source code 1. Sequence alignment (.fas format) of 831 CarD sequences.

### Major dataset

The following datasets were generated:

| Author(s) | Year | Dataset title | Dataset ID and/or URL | Database, license, and accessibility information |
|---|---|---|---|---|
| Chen J, Bae B, Campbell EA, Darst SA | 2015 | Crystal structure of Thermus thermophilus CarD in complex with the Thermus aquaticus RNA polymerase beta1 domain | http://www.rcsb.org/pdb/search/structidSearch.do?structureId=4XAX | Publicly available at the RCSB Protein Data Bank (Accession no: 4XAX). |

| Author(s) | Year | Dataset title | Dataset ID and/or URL | Database, license, and accessibility information |
|---|---|---|---|---|
| Bae B, Darst SA | 2015 | Crystal structure of T. aquaticus transcription initiation complex with CarD containing upstream fork promoter | http://www.rcsb.org/pdb/search/structidSearch.do?structureId=4XLS | Publicly available at the RCSB Protein Data Bank (Accession no: 4XLS). |
| Bae B, Darst SA | 2015 | Crystal structure of T. aquaticus transcription initiation complex with CarD containing bubble promoter and RNA | http://www.rcsb.org/pdb/search/structidSearch.do?structureId=4XLR | Publicly available at the RCSB Protein Data Bank (Accession no: 4XLR). |

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
