## [Decision Letter]

Thank you for submitting your work entitled “CarD uses a minor groove wedge mechanism to stabilize the RNA polymerase open promoter complex” for peer review at *eLife*. Your submission has been favorably evaluated by Richard Losick (Senior Editor) and three reviewers, one of whom is a member of our Board of Reviewing Editors.

The following individuals responsible for the peer review of your submission have agreed to reveal their identity: Stephen Harrison (Reviewing Editor and peer reviewer) and Carol Gross (peer reviewer).

The reviewers have discussed the reviews with one another, and the Reviewing Editor has drafted this decision to help you prepare a revised submission.

The structure of an open complex of *Thermus aquaticus* RNAP with the CarD transcription factor (which conveniently diffused into crystals of the *Taq* RPo) shows that its C-terminal domain contacts the upstream fork. A hinge between the CTD and the domain that binds the β1-lobe allows the former to dock into the fork. Various biochemical measurements support the interpretation that CarD increases the RPo lifetime. This work rationalizes divergent structures in the literature: by using disulfide bond formation they validate that *Tth* CarD structure in the open complex reported here, accurately reflects that of the active form of Mycobacterium tuberculosis CarD. The authors further validate the structure by showing that: (a) KMnO_4_ footprinting indicates that DNA structure is unchanged but that the open complex is more stable, (b) CarD has no stimulatory effect on the open bubble construct and (c) the T is preferred at the -12 non-template position, over any other base as surmised from suggested contact between the crucial W86 residue. The authors also explore the effects of substituting other aromatics or branched chain amino acids at the Cα position. That some of this characterization has already been performed with the mycobacterial enzyme provides evidence for the generality of the mode of action of CarD.

Essential revisions:

1) Throughout the study, Bae et al. use a total of three promoters (full con promoter for structural studies with mismatch bubble, 23S promoter and AP3 promoter). It is not always clear why a certain promoter was used for a certain experiment (e.g. Figure 3). The authors make use of the 23S and the AP3 promoter for their biochemical assays, but do not provide any biochemical data for the full con promoter that was used for the structural studies. The authors should explain why the full con promoter (without the mismatch bubble) wasn't used in addition to the 23S and AP3 promoter to validate the structural results. For easier comparison of the different promoters (full con, 23S and AP3), please provide one figure with alignments of the different promoters.

2) The abortive transcription assays (Figures 3, 4, 5 and 6) only show the trinucleotide transcription product. Were there any further run-off products observed due to mismatch incorporation (“leaky transcription”)? We urge the authors to provide the full gel or to state whether they detected mismatch incorporation.

3) To increase clarity of the overall presentation, the authors should provide an additional (supplemental) figure showing a protein sequence alignment of CarD from different organisms, to support the claim that CarD residue W86 is universally conserved.

4) In the same vein, it would be good to see the distribution of CarD. Is it present in all gram positives (+ mycobacteria) and absent in all gram negatives?

5) The authors claim that CarD “does not alter the transcription bubble” by providing data from a KMnO_4_ footprinting assay (Figure 1). The authors should clarify: a potassium permanganate footprinting assay provides information about the general accessibility of a thymine base, but it does not provide information about the CarD dependent conformational differences (e.g. base flipping) of KMnO_4_ accessible thymine bases. The wording should be changed to reflect this distinction.

6) The polymerase element β1-lobe is not defined in the manuscript. Which residues does it contain? Do we understand correctly that this domain corresponds to the eukaryotic RNAP II protrusion domain? Please clarify in the text, because the eukaryotic RNAP II Rpb2 “lobe” flanks the downstream cleft and thus is different from the bacterial “lobe” – the nomenclature can lead to confusion.

7) We recommend that the authors acknowledge that the CarD mechanism could be more complex and affect additional steps in the RPo formation.

8) Is it possible to further probe the W86-T_-12_ (nt) interaction? For example, phenylalanine activates significantly-would it be expected to make the same contact, or to change the base preference? If so, would it be informative to look at other bases in the T_-12_ (nt) position?

9) CarD was originally described as an rRNA regulator in response to nutrient deprivation, mediating the stringent response. Now that we know the mode of action of CarD, it would be good to return to that phenotype, and describe how it might fit in.

10) There is considerable redundancy, both within this paper and with the accompanying paper on RPo. The authors should simply cross-reference and shorten the present manuscript to eliminate internal redundancy.

---

## [Author Response]

*Essential revisions*:

*1) Throughout the study, Bae et al. use a total of three promoters (full con promoter for structural studies with mismatch bubble, 23S promoter and AP3 promoter). It is not always clear why a certain promoter was used for a certain experiment (e.g.*
Figure 3*). The authors make use of the 23S and the AP3 promoter for their biochemical assays, but do not provide any biochemical data for the full con promoter that was used for the structural studies. The authors should explain why the full con promoter (without the mismatch bubble) wasn't used in addition to the 23S and AP3 promoter to validate the structural results. For easier comparison of the different promoters (full con, 23S and AP3), please provide one figure with alignments of the different promoters*.

The full con promoter sequence was derived from an in vitro evolution (SELEX) protocol optimizing for binding to *Eco* RNAP σ^S^-holoenzyme, but subsequent analysis indicated the sequence appeared to be optimized for binding σ^70^-holoenzyme as well (17). Because of the extremely high conservation of primary σ’s (such as *Eco* σ^70^ and *Taq* σ^A^), the sequence is almost certainly optimized for binding *Taq* σ^A^ -holoenzyme as well. We use this sequence for many of our structural studies to achieve high-affinity, homogeneous complexes important for crystallization. The promoter is extremely active in abortive initiation assays but is actually a poor promoter in vitro and in vivo in run-off assays since there is an issue with promoter escape (RNAP is bound too tightly to the promoter). Because of these properties, the promoter is unlikely to be regulated (it’s already as active as it can be in abortive initiation) and we generally don’t use it for transcription assays, etc.

AP3 is a native *Mtb* rRNA promoter – its regulation by *Mtb* CarD has already been extensively studied (44; 9). In order to analyze more than one promoter (i.e. to show that the effects of CarD are not promoter-specific), we also studied 23S, a native *Tth* rRNA promoter. In each promoter-based assay, the effects of CarD on each promoter were qualitatively the same. In general, we present the results from *Tth* 23S since most of the studies used Thermus Eσ^A^ and CarD. In some cases, it was advantageous to use *Mtb* AP3 instead, and the rationale for using *Mtb* AP3 in each of these cases was as follows:

a) For the KMnO_4_ assays (Figure 1), we show the results from the AP3 promoter because there are more thymines in the t-strand DNA within the expected transcription bubble (3 T’s for AP3, only 1 for 23S).

b) We have included a paragraph (end of the Introduction) explaining the use of the three promoter sequences.

c) We have added a sentence explaining the use of *Mtb* AP3 for Figure 3 (in the subsection “Role of a conserved CarD Trp residue in CarD function”): “We used *Mtb* AP3 for this analysis since *Tth* Eσ^A^ was much more active on this promoter than on *Tth* 23S, allowing us to analyze the much weaker activity of the mutant promoters”.

d) For Figure 4 (testing the function of *Mtb* CarD2C under oxidizing and reducing conditions), we used a mycobacterial transcription system (44; 9) so we used the *Mtb* AP3 promoter (see subsection “The Thermus CarD/RNAP initiation complex structures represent the active conformation of CarD”).

We have included an alignment of all three promoters (Figure 1—figure supplement 1).

*2) The abortive transcription assays (*Figures 3, 4, 5 and 6*) only show the trinucleotide transcription product. Were there any further run-off products observed due to mismatch incorporation (“leaky transcription”)? We urge the authors to provide the full gel or to state whether they detected mismatch incorporation*.

Abortive initiation assays were initiated using specific dinucleotide primers in the presence of only the next α-^32^P-NTP (for AP3, 250 µM GpU + 50 µM α-^32^P-UTP; for 23S, 250 µM UpG + 50 µM α-^32^P-CTP, see Materials and methods). The dominant products were the expected products (GpUp*U for AP3, UpGp*C for 23S). We have included the full, annotated gels in the supplementary figures (Figure 3—figure supplement 1, Figure 4—figure supplement 1, Figure 5—figure supplement 1, Figure 5—figure supplement 2, Figure 6—figure supplement 1).

*3) To increase clarity of the overall presentation, the authors should provide an additional (supplemental) figure showing a protein sequence alignment of CarD from different organisms, to support the claim that CarD residue W86 is universally conserved*.

We have previously published a CarD alignment (44) based on 452 sequences that represented the six diverse groups of bacteria in which CarD is found. This paper also showed the phylogenetic distribution of CarD and the frequency of CarD’s occurrence in these groups. To update and extend this analysis, we searched the updated database within each listed phylum (http://blast.ncbi.nlm.nih.gov/Blast.cgi?PAGE_TYPE=BlastSearch&BLAST_SPEC=MicrobialGenomes) and discovered that CarD is even more widely distributed than originally described (Srivastav et. al., 2013), occurring in 11 of 26 bacteria phyla searched. We now include a table documenting our findings (Supplementary file 1). We also include an alignment of CarD from 11 major subgroups, mostly grouped by phylum, but also dividing the proteobacteria in subgroups to show diversity within that phylum

(Figure 2—figure supplement 1). We also include a large data supplemental file (Supplementary data file 1) containing an alignment of 831 CarD sequences.

4) In the same vein, it would be good to see the distribution of CarD. Is it present in all gram positives (+ mycobacteria) and absent in all gram negatives?

Please see response to point 3.

*5) The authors claim that CarD “does not alter the transcription bubble” by providing data from a KMnO*_*4*_
*footprinting assay (*Figure 1*). The authors should clarify: a potassium permanganate footprinting assay provides information about the general accessibility of a thymine base, but it does not provide information about the CarD dependent conformational differences (e.g. base flipping) of KMnO*_*4*_
*accessible thymine bases. The wording should be changed to reflect this distinction*.

The claim that CarD “does not alter the transcription bubble” is primarily based on comparison of the RPo structure (accompanying paper) and the CarD/RPo structure, as illustrated in Figure 1—figure supplement 6, not the KMnO_4_ footprinting data (Figure 1). The KMnO_4_ footprinting data (Figure 1), obtained in solution assays, supports the structural observations (from crystals). We state that: “The KMnO_4_ reactivity of thymine (T) bases within the transcription bubble is identical in the presence or absence of CarD (Figure 1, lanes 2 and 3), supporting the structural observation that the transcription bubble is the same with or without CarD”. This statement does not imply any structural over interpretation of the KMnO_4_ footprinting data. This paragraph (subsection “The CarD-CTD interacts with the upstream ds/ss junction of the transcription bubble”), and this statement in particular, seems sufficiently clear and we have not revised the text significantly here.

*6) The polymerase element β1-lobe is not defined in the manuscript. Which residues does it contain? Do we understand correctly that this domain corresponds to the eukaryotic RNAP II protrusion domain? Please clarify in the text, because the eukaryotic RNAP II Rpb2 “lobe” flanks the downstream cleft and thus is different from the bacterial “lobe” – the nomenclature can lead to confusion*.

We have clarified that the bacterial RNAP β1-lobe corresponds to the eukaryotic RNAP II Rbp2 protrusion domain (in the fourth paragraph of the Introduction) and we have denoted the residues in the legend to Figure 1 (where the β1-lobe is first identified).

*7) We recommend that the authors acknowledge that the CarD mechanism could be more complex and affect additional steps in the RPo formation*.

We agree with the reviewers and have added that CarD could potentially affect other steps in the transcription initiation steps (please see the Discussion).

*8) Is it possible to further probe the W86-T*_*-12*_
*(nt) interaction? For example, phenylalanine activates significantly-would it be expected to make the same contact, or to change the base preference? If so, would it be informative to look at other bases in the T*_*-12*_
*(nt) position?*

We probed the interaction by mutating W86 to other hydrophobic residues and showed that activation is considerably diminished but not abolished. An F (Phe) substitution shows partial activation (∼1.6 fold versus 3 fold by wild-type CarD). We hypothesize this is due to the loss of contact between the Nԑ of W86 and O2 of the thymine base. An F substitution is able to activate better than the branched or smaller hydrophobic residues because it is more similar structurally to a W. We find substituting the -12T with other bases also diminishes activation to the same extent as an F substitution (∼1.25 to 1.7 versus 3.2 fold), supporting our hypothesis that the loss of interaction between W86 and the -12T reduces the fold- activation roughly 2 fold. We have added this observation to the manuscript (in the subsection “Role of a conserved CarD Trp residue in CarD function”).

*9) CarD was originally described as an rRNA regulator in response to nutrient deprivation, mediating the stringent response. Now that we know the mode of action of CarD, it would be good to return to that phenotype, and describe how it might fit in*.

CarD was indeed originally described as a rRNA regulator in response to nutrient deprivation (45). Subsequently, we discovered that CarD is a highly expressed gene that is present at almost all promoters during exponential growth (44) and so we consider it to be part of the basal transcription machinery in mycobacteria. CarD is expressed at even higher levels during nutrient deprivation, oxidative and genotoxic stress (45) – it certainly regulates rRNA transcription in those circumstances, but it ‘regulates’ almost all other genes as well. Previously, we addressed the problems with the initial interpretations of the physiological effects of depleting CarD (CarD is essential so it cannot be knocked out) – because CarD is essential, highly expressed, and regulates almost all promoters, depleting CarD causes pleiotropic effects that make it impossible to ascribe direct, specific effects (44). We have revised the manuscript to clarify this point (in the Introduction).

*10) There is considerable redundancy, both within this paper and with the accompanying paper on RPo. The authors should simply cross-reference and shorten the present manuscript to eliminate internal redundancy*.

We have removed significant redundancy with the accompanying paper, mainly in the Materials and methods.